# Clinical Features and Outcomes of Persistent Candidemia Caused by *Candida albicans* versus Non-*albicans Candida* Species: A Focus on Antifungal Resistance and Follow-Up Blood Cultures

**DOI:** 10.3390/microorganisms11040928

**Published:** 2023-04-03

**Authors:** Shiori Kitaya, Hajime Kanamori, Yukio Katori, Koichi Tokuda

**Affiliations:** 1Department of Infectious Diseases, Internal Medicine, Graduate School of Medicine, Tohoku University, Sendai 980-8574, Japan; tokuda@med.tohoku.ac.jp; 2Department of Otolaryngology, Head and Neck Surgery, Graduate School of Medicine, Tohoku University, Sendai 980-8574, Japan; yukio.katori.d1@tohoku.ac.jp

**Keywords:** hospital-acquired persistent candidemia, antifungal resistance, clearance of persistent candidemia

## Abstract

The clinical distinctions among variations in *Candida* species, antifungal resistance (AFR), and clearance status of hospital-acquired persistent candidemia (HA-PC) remain uncertain. This secondary analysis of a retrospective cohort study aimed to assess the differences in HA-PC based on different *Candida* species, AFR, and persistent candidemia (PC) clearance status. A retrospective review was conducted using medical records from Tohoku University Hospital of patients for whom blood cultures were performed between January 2012 and December 2021. PC cases were categorized into groups based on *Candida* species, azole, or echinocandin resistance, as well as PC-clearance status, and the respective characteristics were analyzed. The HA-PC non-clearance group had a tendency toward higher 30–90-day and 90-day mortality rates compared to the HA-PC-clearance group in both the susceptible and resistant strain groups, with the former group demonstrating a statistically significant difference (odds ratio = 19, *p* = 0.028). The high mortality rate observed in the *Candida* non-*albicans* and resistant strain groups necessitates a more meticulous therapeutic management approach for PC. Follow-up blood cultures and confirmation of PC clearance are useful for improving the survival rates of both the HA-PC-susceptible and -resistant strain groups.

## 1. Introduction

Among hospital-acquired (HA) infections, fungemia is the fourth leading cause of HA-bloodstream infections [1], with the infection rate increasing annually [2,3] and the mortality rates ranging from 50% to 71% [4]. Some of the most representative and clinically important causative species of fungemia are *Candida* species, which account for approximately 97% of HA fungemia [5]. Recently, non-*albicans Candida* (NAC) species, particularly *Candida parapsilosis* (*C. parapsilosis*), have increasingly replaced *Candida albicans* (*C. albicans*) as the causative agent of persistent candidemia (PC) [6,7]. The underlying factors that can positively influence the persistence of blood culture (BC) in candidemia include the baseline characteristics of the host (e.g., cancer, immunosuppressive medication, and neutropenia), antifungal resistance (AFR), the efficacy of therapeutic management, and source control [8], while central venous catheters (CVCs), metastatic infection foci, ineffective empirical treatment, long hospital stays, and severe sepsis were found to be independent risk factors for PC [8,9]. The mortality rate is higher for PC than for non-PC [10]. In our previous study, we observed that the mortality rate of the PC non-clearance group was significantly higher than that of the clearance group [11].

Recently, reports of international epidemics and outbreaks caused by resistant *Candida* strains have highlighted resistant *Candida* strains as being one of the key issues in terms of global infection control [12,13]. Antifungal resistance is less common in *C. albicans* than that in NAC species; however, it has been reported with long-term antifungal use and recurrent infections, such as those with chronic mucocutaneous candidiasis or recurrent oropharyngeal candidiasis in patients with uncontrolled human immunodeficiency virus infections [4]. Other characteristics of *C. albicans* include a greater tendency to form biofilms than NAC species [14]. Biofilm production by *C. albicans* is important for its resistance, with multiple studies reporting up to 1000-fold higher drug resistance in biofilm-forming cells compared with that in non-biofilm cells in vitro [15]. Biofilms have also been reported to be genetically resistant to amphotericin B and fluconazole, not only in vitro but also clinically, providing the organism with protection and the opportunity to withstand high concentrations of antifungal agents [16,17].

AFR is more pronounced in NAC species. As seen with other *Candida* species, certain *C. parapsilosis* isolates are reportedly increasingly resistant to azoles [14,15]. Fluconazole resistance rates in *C. parapsilosis* isolates were 5-fold higher than those in *C. albicans* [16]. With regards to echinocandin resistance, *C. parapsilosis* has a unique intrinsic resistance to these drugs; however, repeated exposure to echinocandins is a risk factor for *C. parapsilosis* resistance [17]. In *Candida glabrata* (*C. glabrata*) fungemia, 30-day and 90-day mortality rates were significantly higher in patients with fluconazole-resistant isolates compared to those in patients with fluconazole-susceptible dose-dependent isolates [18].

Although several studies on HA-PC have been conducted in recent years, not enough research has been conducted to investigate the differences in clinical outcomes between different *Candida* species, with and without AFR. Additionally, there are no reports comparing the clinical outcomes of HA-PC-resistant and -susceptible strains in terms of PC clearance. Therefore, this study aimed to investigate the clinical outcomes and mortality rates associated with HA-PC, focusing on *Candida* species, AFR, and PC clearance.

## 2. Methods

### 2.1. Study Design and Setting

This is a secondary analysis study that uses electronic medical records and hospital records of Tohoku University Hospital from January 2012 to December 2021 [11]. The data for this study were extracted from the computerized records of the Department of Laboratory Medicine and the medical records and database of the Department of Infectious Diseases. The clinical data included demographic information (age, sex, comorbidities, body mass index, and body temperature), blood test results (levels of serum white blood cells, neutrophils, and C-reactive protein), and clinical characteristics (presence and removal of intravascular devices, history of cardiovascular surgery, use of extracorporeal membrane oxygenation, continuous hemodiafiltration, and mechanical ventilation, interval between initial BC and follow-up blood culture (FUBC), duration of candidemia, site of infections, duration of hospitalization, time spent in the intensive care unit, high care unit, or coronary care unit, antifungal drug use, performance of source control, and early (30-day), late (30–90-day), and 90-day mortality).

All patients diagnosed with candidemia (defined as one or more positive BCs obtained to rule out infection) were included in the study. The exclusion criteria included HA-PC patients under 18 years of age and those with polymicrobial PC. This study was approved by the Human Ethical and Clinical Trial Committee of Tohoku University Hospital (approval: 2019-1-270). The requirement for patient consent was waived because of the retrospective nature of the study.

### 2.2. Definitions and Outcomes

The definitions of PC, FUBC, source of PC, PC duration, PC clearance, source control, neutropenia, immunosuppression, BC collection, adequacy of antimicrobial therapy, and patient selection algorithm were determined in accordance with our previous report [11]. HA-PC was defined as cases in which the same *Candida* species were detected on two or more consecutive occasions from BCs collected >48 h after admission [19]. The intravascular devices that were considered were conventional CVCs, peripherally inserted central catheters, tunneled CVCs, and implanted central venous ports. The cardiovascular surgical procedures included valve replacement, vascular graft replacement, ventricular assist device implantation, and cardiac device implantation. Endovascular device infections encompassed those of vascular grafts and left ventricular assist devices. This study identified the following *Candida* strains as causative agents of PC other than *C. albicans*: *C. parapsilosis*, *C. tropicalis*, *C. guilliermondii*, *C. glabrata*, *C. krusei*, *C. lusitaniae*, and *C. famata*. *Candida* spp. were identified using a VITEK MS system (bioMérieux, Marcy-l’Etoile, France), and spectra were interpreted using their MYLA^®^ microbiology middleware. The RAISUS S4 system (Nissui Pharmaceutical Co., Ltd., Tokyo, Japan) was used for susceptibility testing, and the Clinical and Laboratory Standards Institute (CLSI) criteria were used for interpreting the susceptibility results [20]. The primary outcome variables in this study were early (30-day), late (30–90-day), and 90-day mortality after the initial BC. Early (30-day), late (30–90-day), and 90-day mortality were defined by death due to PC, respectively. Interpretive breakpoints for azole or echinocandin were based on criteria recommended by the CLSI [20]. Resistant *Candida* strains were defined as those resistant or susceptible in a dose-dependent manner to one or both of azole or echinocandin.

### 2.3. Statistical Analysis

The results were expressed as the median value with a 95% confidence interval or as a proportion of the total number of patients or isolates. We used the Mann–Whitney U test to compare the averages of continuous variables and Fisher’s exact test to compare the proportions of categorical variables between the two groups. The analysis was performed using the JMP Pro 16 statistical analysis software (SAS Institute, 2021). Differences were considered significant at a corrected *p*-value of <0.05.

## 3. Results

### 3.1. Clinical Characteristics of Hospital-Acquired Persistent Candidemia Caused by Candida albicans and Non-albicans Candida

The clinical characteristics of HA-PC caused by *C. albicans* and NAC are described in detail below and shown in Table 1. The body temperature, white blood cell count, and neutrophil count were significantly higher in the HA-persistent *C. albicans* fungemia group than in the HA-persistent non-*albicans* candidemia group (*p* = 0.003, *p* < 0.001, and *p* < 0.001, respectively). The ratio of immunosuppression and catheter-related bloodstream infection (CRBSI) was significantly higher in the HA-persistent non-*albicans* candidemia group than in the HA-persistent *C. albicans* fungemia group (Odds ratio (OR) = 6.3, *p* = 0.026, and OR = 4.6, *p* = 0.004, respectively). Although there were no statistically significant differences, the HA-persistent non-*albicans* candidemia group tended to have higher early (30-day), late (30–90-day), and 90-day mortality ratios than the HA-persistent *C. albicans* fungemia group.

Data are presented as numbers (%) unless indicated otherwise. In the table, *p*-values are listed only for items that show significant differences. The blood test was performed on the same day as the blood culture collection. Immunosuppression was considered in neutropenia, hematopoietic stem-cell transplantation, solid organ transplantation, and corticosteroid therapy (prednisone 16 mg per day for 15 days). Cardiovascular surgery includes valve replacement, vascular graft replacement, ventricular assist device, and cardiac device implantation. Endovascular device infections encompass those of vascular grafts and left ventricular assist devices. Hospital-acquired persistent candidemia was defined as cases in which the same *Candida* species were detected on two or more consecutive occasions from blood cultures collected >48 h after admission. The persistent non-*albicans* candidemia group includes persistent candidemia caused by *Candida parapsilosis* (*n* = 20), *Candida tropicalis* (*n* = 3), *Candida guilliermondii* (*n* = 2), *Candida glabrata* (*n* = 5), *Candida krusei* (*n* = 1), *Candida lusitaniae* (*n* = 1), and *Candida famata* (*n* = 1). BMI, body mass index; CCU, coronary care unit; CI, confidence interval; CRBSI, catheter-related bloodstream infection; ECMO, extracorporeal membrane oxygenation; ESDR, end-stage renal disease; FUBC, follow-up blood culture; HCU, high care unit; ICU, intensive care unit; IQR, interquartile range.

### 3.2. Clinical Characteristics of Hospital-Acquired Persistent Candidemia in Terms of Azole or Echinocandin Resistance and Clearance of Persistent Candidemia

The clinical characteristics of HA-PC resistant to azole or echinocandin and susceptible strain groups are shown in Table 2. Significant differences were observed in vital signs and laboratory markers between the HA-PC-resistant and -susceptible strain groups, with body temperature, white blood cell count, and neutrophil count being significantly higher in the former group than in the latter group (*p* = 0.015, *p* = 0.010, and *p* = 0.006, respectively). Regarding the site of infection, CRBSI was significantly higher in the HA-PC-resistant strain group than in the HA-PC-susceptible strain group (OR = 4.2, *p* = 0.008). Intraocular candidiasis was significantly lower in the HA-PC-resistant strain group than in the HA-PC-susceptible strain group (*p* = 0.020). Although there were no statistically significant differences, the persistent HA-PC-resistant strain group tended to have higher early (30-day), late (30–90-day), and 90-day mortality ratios than the susceptible strain group.

Data are presented as numbers (%) unless indicated otherwise. In the table, *p*-values are listed only for items that show significant differences. The blood test was performed on the same day as the blood culture collection. Immunosuppression was considered in neutropenia, hematopoietic stem-cell transplantation, solid organ transplantation, and corticosteroid therapy (prednisone 16 mg per day for 15 days). Cardiovascular surgery includes valve replacement, vascular graft replacement, ventricular assist device, and cardiac device implantation. Endovascular device infections encompass those of vascular grafts and left ventricular assist devices. HA-PC was defined as cases in which the same *Candida* species were detected on two or more consecutive occasions from blood cultures collected >48 h after admission. Resistant *Candida* strains were defined as those strains resistant or susceptible in a dose-dependent manner to one or both of azole or echinocandin. BMI, body mass index; CCU, coronary care unit; CI, confidence interval; CRBSI, catheter-related bloodstream infection; ECMO, extracorporeal membrane oxygenation; ESDR, end-stage renal disease; FUBC, follow-up blood culture; HA-PC, hospital-acquired persistent candidemia; HCU, high care unit; ICU, intensive care unit; IQR, interquartile range.

The clinical characteristics of HA-PC-resistant and -susceptible strain groups in terms of PC clearance are shown in Appendix A. The median patient age was significantly higher in the HA-PC-resistant strain PC non-clearance group than in the HA-PC-resistant strain PC-clearance group (*p* = 0.025). The late (30–90-day) and 90-day mortality rates were approximately 19 times higher in the HA-PC-susceptible strain PC non-clearance group than in the HA-PC-susceptible strain PC-clearance group (both OR = 19 and *p* = 0.028). For the HA-PC-resistant strain groups, mortality tended to be higher in the HA-PC PC non-clearance group than in the HA-PC PC-clearance group; the intergroup difference was not significant.

## 4. Discussion

### 4.1. Differences in Clinical Characteristics between Hospital-Acquired Persistent Candidemia Caused by Candida albicans and Non-albicans Candida

In this study, the early (30-day), late (30–90-day), and 90-day mortality rates in the HA-persistent non-*albicans* candidemia group were higher than those of the HA-persistent *C. albicans* fungemia group. Regarding virulence and pathogenicity, certain NAC species exhibit comparable or greater virulence in humans compared to *C. albicans* [21]. The mortality rate associated with NAC infections is species-dependent, with *C. tropicalis* and *C. glabrata* reportedly having the highest mortality rates ranging from 40% to 70% [21]. The fatality rate for each *Candida* species in this study was as follows: *C. albicans* 3/27 cases (11.1%), NAC; *C. parapsilosis* 5/20 cases (25%), *C. glabrata* 1/2 cases (50%), *C. tropicalis* 2/3 cases (66.7%), and *C. guilliermondii* 2/2 cases (100%). The prevalence of antifungal-resistant strains was higher among the NAC group than among the *C. albicans* group (28/33 cases (84.8%) vs. 6/27 cases (22.2%)). Progressive resistance of *Candida* to azoles and echinocandins may be linked to the heightened mortality rate associated with invasive candidiasis [22]. These results imply that the high virulence of NAC and the high incidence of antifungal-resistant strains may have contributed to the elevated mortality rates of HA-persistent non-*albicans* candidemia observed in this study. In CRBSI, the most frequent distant metastasis in the present study, intraocular candidiasis, and septic embolism were complications reported in 8 of 15 patients (53.3%) when *C. albicans* was the causative pathogen, although no cases were reported with NAC as the causative organism. *C. albicans* accounts for approximately 50% of hematogenous disseminated candidiasis cases, followed by *C. glabrata*, *C. parapsilosis*, and *C. tropicalis*, each comprising approximately 10–25% of cases [23]. It was possible that some of the cases of HA-persistent non-*albicans* candidemia in this study were not thoroughly investigated for distant plexus metastasis due to an unfavorable health status or best supportive care policy. However, the high incidence of distant plexus complications in CRBSI caused by *C. albicans* in this study suggests that the pathogen has a proclivity for forming such distant metastatic foci. Conversely, in the case of HA-persistent non-*albicans* candidemia, it is important to note that the mortality rate is high even when there is no complication of disseminated lesions.

### 4.2. Differences in Clinical Characteristics in Terms of Azole or Echinocandin Resistance and Clearance of Persistent Candidemia

Regarding the site of infection, CRBSI was significantly higher in the HA-PC-resistant strain group than that in the HA-PC-susceptible strain group. However, the presence of intraocular candidiasis was significantly lower in the HA-PC-resistant strain group than that in the HA-PC-susceptible strain group. *C. parapsilosis* and *C. albicans* are the predominant etiological agents responsible for CRBSI resulting from candidiasis [24,25]. In the present study, *C. albicans* was the most common causative agent of CRBSI (15/40; 37.5%), followed by *C. parapsilosis* (14/40; 35%); in the case of PC, the same species as those reported previously were found to be common. Furthermore, the prevalence of antifungal-resistant *C. parapsilosis* strains causing PC was observed to be 95% (19/20; 95%), and all *C. parapsilosis* instances responsible for CRBSI were antifungal-resistant, i.e., 14 of the 14 cases (100%). Consequently, the elevated AFR rate among *C. parapsilosis* strains as causative agents of CRBSI might have contributed to the high incidence of CRBSI in the HA-PC-resistant strain group. The mortality rate among patients diagnosed as having CRBSI appeared to be higher in the resistant strain group than that in the susceptible strain group (9/27 cases (33.3%) vs. 1/13 cases (7.7%)). The proportion of cases with successful source control was comparable in both the susceptible and resistant strain groups, regardless of whether the outcome was survival or death. Specifically, in the susceptible strain group, 11 of 12 cases (91.7%) demonstrated adequate source control for survivors and 1 of 1 case (100%) for deaths. In the resistant strain group, 17 of 18 cases (94.4%) exhibited adequate source control for survivors and 7 of 9 cases (77.8%) for deaths. The incidence of fungal CRBSI has been established as correlating with recurrent catheter-associated infections, even with the removal of the infected catheter and the implementation of appropriate antifungal treatment [26], thereby presenting a challenge in the effective management of fungal CRBSI. Regarding the resistant strain group, which had a high mortality rate, the mortality rate for cases with no confirmed clearance of PC was 6 of 12 cases (50%), compared with 3 of 15 cases (20%) for those with confirmed clearance of PC. This suggests that in addition to appropriate source control, confirming the clearance of PC may be useful to improve survival in the case of PC caused by CRBSI.

The American Academy of Ophthalmology does not advocate for a standard ophthalmologic evaluation upon laboratory detection of systemic candidemia; however, an ophthalmologic evaluation is deemed a prudent course of action for a patient presenting with signs indicative of ocular infection, regardless of *Candida* septicemia [27]. The incidence of endogenous fungal endophthalmitis and choroiditis is generally <5%, with a declining trend [28,29,30]. This trend is considered to be due to several factors, including improvements in antifungal therapy, prompt initiation of treatment, and prophylactic treatment when clinical suspicion is high [31]. In the present study, 5 of 60 (8.3%) patients with PC had intraocular candidiasis, which surpasses the previously reported rate [28,29,30]. This may possibly be because of the fact that PC increases the chance of infection by increasing the eye exposure time to *Candida* via capillary blood flow; a previous study has reported that PC is a risk factor for intraocular candidiasis [32]. Among patients who developed candidemia and had abnormalities suggestive of *Candida* eye involvement, most (6/7; 85.7%) were asymptomatic [33]. Similarly, in the present study, most of the patients (4/5; 80%) were asymptomatic as far as the medical records were concerned. Therefore, the results of this study suggest the need for more aggressive ophthalmologic evaluation in cases of PC, even in the absence of ocular symptoms. The causative species of intraocular candidiasis in this study was *C. albicans* for all cases; this is consistent with the results of previous studies [32,33]. Candidemia with *C. albicans* is a risk factor for *Candida* chorioretinitis [34,35]. *C. albicans*, the most common causative organism in this study, was more common in the susceptible strains (21/27; 77.8%). This may be why intraocular candidiasis was more common in the HA-PC-susceptible strain group.

We found that in both the HA-PC-susceptible and -resistant strain groups, the HA-PC non-clearance group had a higher mortality rate than the HA-PC-clearance group. The most common cause of failure to confirm HA-PC clearance was the lack of confirmation of negative BCs owing to the best supportive care policy, e.g., lung adenocarcinoma, malignant lymphoma, and acute interstitial pneumonia (7/21; 33.3%). The next most common causes of failure to confirm HA-PC clearance was the unknown focus of infection (5/21; 23.8%), insufficient source control (2/21; 9.5%), the death of a patient from a disease other than an infectious disease such as cerebral hemorrhage and liver failure (2/21; 9.5%), and patient death before FUBC was performed (2/21; 9.5%). Despite its limitations, our results suggest that in HA-PC patients undergoing aggressive close examination and treatment, performing FUBC to confirm clearance of PC may contribute to improved survival, regardless of the presence or absence of azole or echinocandin resistance.

## 5. Conclusions

To the best of our knowledge, this is the first study investigating the characteristics of HA-PC in the context of differences among *Candida* species, azole or echinocandin resistance, and clearance of PC. Our study has several limitations. First is the difficulty in generalizing the findings from a single center to other settings. Second is the small number of cases of HA-PC-susceptible strains in the non-clearance group. Thirdly, we were unable to identify *Candida auris* because we did not adopt the latest version (4.14 or later) of VITEK MS at our institution. Despite its limitations, this is a long-term, decadal HA-PC study, reporting clinically significant findings. Patients with HA-PC caused by *Candida* non-*albicans* or azole- or echinocandin-resistant strains tend to have higher mortality rates and require particularly careful therapeutic management. Confirming PC clearance via FUBC can help improve the survival rate of patients infected with HA-PC-susceptible and resistant strains.

## Figures and Tables

**Table 1 microorganisms-11-00928-t001:** Clinical characteristics of hospital-acquired persistent candidemia caused by *Candida albicans* and non-albicans *Candida*.

	Persistent Non-*albicans* Candidemia Group (*n* = 33)	Persistent *Candida albicans* Fungemia Group (*n* = 27)	Odds Ratio [95% CI]	*p*-Value
**Demography**				
Sex (male, %)	19 (57.6)	19 (70.4)	0.6 [0.2, 1.7]	
Age, years, median (IQR)	62.5 (52.0–70.5)	64.0 (51.0–73.0)		
**Comorbidities**				
Diabetes mellitus	3 (9.1)	7 (25.9)	0.3 [0.1, 1.2]	
ESDR on hemodialysis	0 (0)	1 (3.7)	0	
Liver cirrhosis	2 (6.1)	1 (3.7)	1.7 [0.1, 19.6]	
Solid malignancy	11 (33.3)	16 (59.3)	0.3 [0.1, 1]	
Hematologic malignancy	4 (12.1)	0 (0)	–	
Neutropenia	5 (15.2)	0 (0)	–	
Immunosuppression	11 (33.3)	2 (7.4)	6.3 [1.2, 31.3]	0.026
**Vital signs**				
BMI, kg/m^2^, median (IQR)	19.4 (17.2–23.9)	20.2 (17.3–21.7)		
Body temperature, °C, median (IQR)	38.0 (37.1–38.8) (*n* = 30)	38.9 (38.1–39.5) (*n* = 24)		0.003
**Laboratory markers**				
White blood cell count, 10⁹/L, median (IQR)	6000.0 (3400.0–10,100.0)	11,600.0 (7800.0–15,250.0)		<0.001
Neutrophil count, 10⁹/L, median (IQR)	5100.0 (2030.0–9390.0)	10,315.0 (6292.5–13,267.5) (*n* = 26)		<0.001
C-reactive protein, mg/dL, median (IQR)	9.0 (2.2–11.6)	7.7 (4.6–9.5)		
**Devices**				
Intravascular device	28 (84.8)	20 (74.1)	2 [0.5, 7]	
Intravascular device removal	24 (85.7)	18 (90.0)	0.7 [0.1, 4]	
Cardiovascular surgery	4 (12.1)	4 (14.8)	0.8 [0.2, 3.5]	
ECMO	3 (9.1)	1 (3.7)	2.6 [0.3, 26.5]	
Continuous hemodiafiltration	8 (24.2)	4 (14.8)	1.8 [0.5, 6.9]	
Mechanical ventilation	11 (33.3)	7 (25.9)	1.4 [0.5, 4.4]	
**Status of persistent candidemia**				
The period until FUBC is carried out, median (IQR)	4.0 (2.0–6.0)	3.0 (2.0–5.0)		
Duration of candidemia, median (IQR)	6.0 (4.0–11.0)	5.0 (3.0–7.0)		
**Site of infection**				
CRBSI	25 (75.8)	15 (40.5)	4.6 [1.6, 12.9]	0.004
Endovascular devices infections	1 (3.0)	2 (5.4)	0.5 [0, 6.3]	
Septic embolism	0 (0)	1 (2.7)	0	
Thrombophlebitis	0 (0)	4 (10.8)	0	
Infected aneurysm	0 (0)	1 (2.7)	0	
Intraocular candidiasis	0 (0)	5 (13.5)	0	
Skin and soft tissue infections	1 (3.0)	0 (0)	–	
Abscess	1 (3.0)	2 (5.4)	0.5 [0, 6.3]	
Intra-abdominal infections	0 (0)	1 (2.7)	0	
Urinary tract infections	0 (0)	1 (2.7)	0	
Unknown	5 (15.2)	5 (13.5)	1.1 [0.3, 4.4]	
**Hospital stays**				
Duration of hospitalization, days, median (IQR)	59.5 (42.5–102.8)	88.0 (58.5–117.0)		
Presence of ICU	12 (36.4)	15 (55.6)	0.5 [0.2, 1.3]	
Duration of ICU stay, days, median (IQR)	0 (0–17)	1.0 (0–15.5)		
Presence of HCU	2 (6.1)	3 (11.1)	0.5 [0.1, 3.3]	
Duration of HCU stay, days, median (IQR)	0 (0–0)	0 (0–0)		
Presence of CCU	1 (3.0)	0 (0)	–	
Duration of CCU stay, days, median (IQR)	0 (0–0)	0 (0–0)		
**Intervention**				
The use of antibiotics (Appropriate)	30 (90.9)	27 (100)	0	
Source control	24 (72.7)	15 (55.6)	2.1 [0.7, 6.3]	
**Mortality**				
Early (30-day) mortality	1 (3.0)	0 (0)	–	
Late (30–90-day) mortality	9 (27.3)	3 (11.1)	3 [0.7, 12.5]	
90-day mortality	10 (30.3)	3 (11.1)	3.5 [0.8, 14.3]	

**Table 2 microorganisms-11-00928-t002:** Clinical characteristics of hospital-acquired persistent candidemia in terms of azole or echinocandin resistance.

	HA-PC-Resistant Strain Group (*n* = 34)	HA-PC-Susceptible Strain Group (*n* = 26)	Odds Ratio [95% CI]	*p*-Value
**Demography**				
Sex (male, %)	19 (55.9)	19 (73.1)	0.5 [0.2, 1.4]	
Age, years, median (IQR)	62.5 (52.0–68.0)	66.0 (50.0–74.8)		
**Comorbidities**				
Diabetes mellitus	4 (11.8)	6 (23.1)	0.4 [0.1, 1.8]	
ESDR on hemodialysis	0 (0)	1 (3.8)	0	
Liver cirrhosis	3 (8.8)	0 (0)	–	
Solid malignancy	12 (35.3)	15 (57.7)	0.4 [0.1, 1.1]	
Hematologic malignancy	4 (11.8)	0 (0)	–	
Neutropenia	5 (14.7)	0 (0)	–	
Immunosuppression	10 (29.4)	3 (11.5)	3.2 [0.8, 13.1]	
**Vital signs**				
BMI, kg/m^2^, median (IQR)	19.0 (17.3–22.9)	20.3 (17.2–23.0)		
Body temperature, °C, median (IQR)	38.1 (37.2–38.8) (*n* = 31)	38.9 (38.3–39.6) (*n* = 23)		0.015
**Laboratory markers**				
White blood cell count, 10⁹/L, median (IQR)	6350.0 (3775.0–11,450.0)	10,050.0 (6950.0–15,550.0)		0.010
Neutrophil count, 10⁹/L, median (IQR)	5200.0 (2210.0–9915.0)	8770.0 (5890.0–13,600.0) (*n* = 25)		0.006
C-reactive protein, mg/dL, median (IQR)	8.1 (2.3–11.6)	8.3 (4.9–10.7)		
**Devices**				
Intravascular device	30 (88.2)	18 (69.2)	3.3 [0.9, 12.7]	
Intravascular device removal	26 (86.7)	16 (88.9)	0.8 [0.1, 5]	
Cardiovascular surgery	6 (17.6)	2 (7.7)	2.6 [0.5, 13.9]	
ECMO	3 (8.8)	1 (3.8)	2.4 [0.2, 24.7]	
Continuous hemodiafiltration	8 (23.5)	4 (15.4)	1.7 [0.4, 6.4]	
Mechanical ventilation	9 (26.5)	9 (34.6)	0.7 [0.2, 2.1]	
**Status of persistent candidemia**				
The period until FUBC is carried out, median (IQR)	4.0 (2.0–6.0)	3.0 (2.0–5.0)		
Duration of candidemia, median (IQR)	6.0 (3.3–10.8)	5.0 (3.3–7.0)		
**Site of infection**				
CRBSI	27 (73.0)	13 (39.4)	4.2 [1.5, 11.4]	0.008
Endovascular devices infections	2 (5.4)	1 (3.0)	1.8 [0.2, 21.1]	
Septic embolism	0 (0)	1 (3.0)	0	
Thrombophlebitis	2 (5.4)	2 (6.1)	0.9 [0.1, 6.7]	
Infected aneurysm	0 (0)	1 (3.0)	0	
Intraocular candidiasis	0 (0)	5 (15.2)	0	0.020
Skin and soft tissue infections	1 (2.7)	0 (0)	–	
Abscess	2 (5.4)	1 (3.0)	1.8 [0.2, 21.1]	
Intra-abdominal infections	0 (0)	1 (3.0)	0	
Urinary tract infections	0 (0)	1 (3.0)	0	
Unknown	3 (8.1)	7 (21.2)	0.3 [0, 1.4]	
**Hospital stays**				
Duration of hospitalization, days, median (IQR)	60.0 (45.0–106.0)	89.0 (45.5–121.5)		
Presence of ICU	12 (35.3)	15 (57.7)	0.4 [0.1, 1.1]	
Duration of ICU stay, days, median (IQR)	0 (0–20.0)	1.0 (0–11.8)		
Presence of HCU	2 (5.9)	3 (11.5)	0.5 [0.1, 3.1]	
Duration of HCU stay, days, median (IQR)	0 (0–0)	0 (0–0)		
Presence of CCU	1 (2.9)	0 (0)	–	
Duration of CCU stay, days, median (IQR)	0 (0–0)	0 (0–0)		
**Intervention**				
The use of antibiotics (Appropriate)	31 (91.2)	26 (100)	0	
Source control	26 (76.5)	13 (50.0)	3.3 [1.1, 9.8]	
**Mortality**				
Early (30-day) mortality	1 (2.9)	0 (0)	–	
Late (30–90-day) mortality	8 (23.5)	4 (15.4)	1.7 [0.4, 6.4]	
90-day mortality	9 (26.5)	4 (15.4)	2 [0.5, 7.3]	

## Data Availability

The datasets created and analyzed during the current study are not publicly available due to them containing a great deal of detailed patient information. The dataset is owned by the Department of Infectious Diseases, Internal Medicine, Tohoku University Graduate School.

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
