# Peer review of "Clinical Features and Outcomes of Persistent Candidemia Caused by Candida albicans versus Non-albicans Candida Species: A Focus on Antifungal Resistance and Follow-Up Blood Cultures"

_microorganisms, 2023, doi:10.3390/microorganisms11040928_

Round 1

Reviewer 1 Report

The authors present an article as a secondary study analysis of data from a hospital on fungemia in patients.  They attempt to show that persistent candidemia caused by species other than C. albicans leads to poor outcomes.  It is an extension of their previous study: “Clinical and Epidemiological Characteristics of Persistent Bacteremia: A Decadal Observational Study”.  This current study seems to include all of the same data but includes a breakdown by species and resistance.

The authors use the term bacteriemia in several locations but isn’t the correct term candidemia as they mentioned in the paper? The reviewer would also suggest the usage of the term fungemia as well. Bacteremia seems incorrect as no microorganisms discussed in this paper are bacteria.

Could the authors confirm: did their test kits include the ability to test for C. auris ? It appears that not cases were found with this species, correct?

Author Response

Reviewer 1

  1. The authors use the term bacteriemia in several locations but isn’t the correct term candidemia as they mentioned in the paper? The reviewer would also suggest the usage of the term fungemia as well. Bacteremia seems incorrect as no microorganisms discussed in this paper are bacteria.

Response: I express my profound gratitude for the insightful observations made by the reviewer. As highlighted by the reviewer, Candida is a type of fungus, thereby making the usage of the term 'bacteremia' incongruous. In light of this, we have diligently revised the text by substituting the term 'bacteremia' with either 'candidemia' or 'fungemia'.

  1. Could the authors confirm: did their test kits include the ability to test for C. auris? It appears that not cases were found with this species, correct?

Response: We extend our sincere gratitude to the reviewer for their invaluable insights. As pointed out by the reviewer, we were unable to identify Candida auris due to not adopting the latest version (4.14 or later) of VITEK MS at our institution (https://www.cdc.gov/fungal/candida-auris/identification.html). However, it is important to note that this study is an analysis of previously collected data in a retrospective cohort design. Moreover, as this study focused on the comparison between the Candida albicans group and non-albicans Candida group, even if Candida auris had been accurately identified, it is believed that the impact on the results (including the tables) would have been minimal, with only slight changes in the species composition within the persistent non-albicans candidemia group. Therefore, we have added the following limitation regarding the inability to detect Candida auris in this study.

“Thirdly, we were unable to identify Candida auris because we did not adopt the latest version (4.14 or later) of VITEK MS at our institution.” (Lines 306–308)

Reviewer 2 Report

The authors present a retrospective review was conducted using medical records from Tohoku University Hospital of patients for whom blood cultures were performed between January 2012 and December 2021. PC cases were categorized into groups based on Candida species, azole, or echinocandin resistance. The paper is interesting but there are some remarks to address

Major revision

Methods: the atuhors declare “…VITEK MS System (bioMérieux, Marcy-l’Etoile, France). A WalkAway 96 plus system (Siemens Healthcare Diagnostics, Deerfield, IL, USA) was used for susceptibility testing”

The version of the software used for the interpetation of spectrum of Candida has to be added.

Unfortunately, VITEK MS system are not able to correctly identify C.auris, and considering the resistance trend observed in data set presented we cannot exclude that the resistance was dued to such pathogens. I suggest to produce an additional test (using ATCC) demostrating that the VITEK MS used is able to correctly identify C.auris.

Considering the AST,  the walkaway used in this work does not reprresent a refenced methods. ASTs have to be performed by microdilution method according to CLSI or EUCAST standards.

Please add in the methods section the standard used for the interpretation: CLSI or EUCAST (they differ significantly)

Considering the above observations it is also important to add in the discussion these possible biases/limitations in the interpretation of the ID and the AST (hence of the reported resistance trend)

Minor revison: please change susceptive in susceptible

Author Response

Reviewer 2

Major revision

  1. The version of the software used for the interpetation of spectrum of Candida has to be added.

Response: We extend our sincere appreciation to the reviewers for their insightful comments. We consider it crucial to provide a detailed account of the software utilized for the spectral analysis of Candida. As such, we have incorporated the ensuing information into the manuscript.

Candida spp. were identified using a VITEK MS system (bioMérieux, Marcy-l'Etoile, France), and spectra were interpreted using their MYLA® microbiology middleware.” (Lines 109–111)

  1. Unfortunately, VITEK MS system are not able to correctly identify C.auris, and considering the resistance trend observed in data set presented we cannot exclude that the resistance was dued to such pathogens. I suggest to produce an additional test (using ATCC) demostrating that the VITEK MS used is able to correctly identify C.auris.

Response: We express our profound appreciation for the insightful recommendations put forward by the reviewer concerning our manuscript. As the reviewer pointed out, we were unable to identify Candida auris because we did not use the latest version (4.14 or later) of VITEK MS at our institution (https://www.cdc.gov/fungal/candida-auris/identification.html). However, it is crucial to acknowledge that this study is a retrospective cohort analysis of preexisting data. Furthermore, since our study mainly focused on comparing the Candida albicans group and non-albicans Candida group, even if Candida auris had been identified accurately, it would have had minimal impact on the results (including tables), with only minor alterations in the species composition within the persistent non-albicans candidemia group. Consequently, we have added a limitation statement that acknowledges the inability to detect Candida auris in this study.

“Thirdly, we were unable to identify Candida auris because we did not adopt the latest version (4.14 or later) of VITEK MS at our institution.” (Lines 306–308)

  1. Considering the AST, the walkaway used in this work does not reprresent a refenced methods. ASTs have to be performed by microdilution method according to CLSI or EUCAST standards.

Please add in the methods section the standard used for the interpretation: CLSI or EUCAST (they differ significantly)

Response: I extend my sincere appreciation to the reviewer for the perceptive observation. The evaluation of Candida susceptibility in our medical institution was carried out employing RAISUS S4 system, in lieu of WalkAway 96 plus system. The AST outcomes were evaluated utilizing the established standards put forth by the CLSI, a fact that has been duly mentioned in our manuscript.

“The RAISUS S4 system (Nissui Pharmaceutical Co., Ltd, Tokyo, Japan) was used for susceptibility testing, and the Clinical and Laboratory Standards Institute (CLSI) criteria were used for interpreting the susceptibility results [20].” (Lines 112–115)

  1. Considering the above observations it is also important to add in the discussion these possible biases/limitations in the interpretation of the ID and the AST (hence of the reported resistance trend)

Response: We express our heartfelt gratitude to the reviewers for their perspicacious remarks. We acknowledge a constraint in this study concerning the construal of AD and AST, specifically, the incapacity to discern Candida auris. As a result, we have incorporated the subsequent declaration in the Limitations section, in lieu of the Discussion section, for the purpose of lucidity and coherence with the manuscript's context.

“Thirdly, we were unable to identify Candida auris because we did not adopt the latest version (4.14 or later) of VITEK MS at our institution.” (Lines 306–308)

Round 2

Reviewer 2 Report

The paper has been sufficiently reviewed  and the limitations described in the conclusions